# The Influence of Sexual Behavior and Demographic Characteristics in the Expression of HPV-Related Biomarkers in a Colposcopy Population of Reproductive Age Greek Women

**DOI:** 10.3390/biology10080713

**Published:** 2021-07-26

**Authors:** George Valasoulis, Abraham Pouliakis, Georgios Michail, Athina-Ioanna Daponte, Georgios Galazios, Ioannis G. Panayiotides, Alexandros Daponte

**Affiliations:** 1Department of Obstetrics & Gynaecology, University Hospital of Larisa, 41334 Larisa, Greece; atdaponte@uth.gr (A.-I.D.); daponte@med.uth.gr (A.D.); 2Hellenic National Public Health Organization—ECDC, 15123 Athens, Greece; 3Second Department of Pathology, National and Kapodistrian University of Athens, Attikon University Hospital, 12462 Athens, Greece; apouliak@med.uoa.gr (A.P.); ioagpan@med.uoa.gr (I.G.P.); 4Department of Obstetrics & Gynaecology, University Hospital of Patras, 26504 Patras, Greece; gmichail@upatras.gr; 5Department of Obstetrics & Gynaecology, University Hospital of Alexandroupolis, 68100 Alexandroupolis, Greece; ggalaz@med.duth.gr

**Keywords:** human papillomavirus, HPV, HPV DNA, mRNA E6 and E7, biomarkers, condoms, CIN, colposcopy, lifestyle

## Abstract

**Simple Summary:**

This observational study aimed to investigate the possible influence of sexual behavior and demographic characteristics in the expression of HPV-related biomarkers in a colposcopy population. Lifestyle factors that impacted HPV DNA positivity in a statistically significant manner were younger age at first sexual intercourse, a history of more than four sexual partners as well as a partner change during the last year before HPV DNA assessment. Although ambitious, the development and validation of lifestyle scoring systems that combine molecular and epidemiological patient data to effectively predict underlying cervical pathology will represent a milestone in the quest of cervical cancer prevention.

**Abstract:**

Despite the significant scientific evolution in primary and secondary cervical cancer prevention in the battle started by George Papanicolaou in the previous century, global cervical cancer mortality rates remain disappointing. The widespread implementation of HPV-related molecular markers has paved the way to tremendous developments in cervical cancer screening, with the transition from cytological approach to the more accurate and cost-effective HPV testing modalities. However, the academic audience and different health systems have not yet adopted a universal approach in screening strategies, and even artificial intelligence modalities have been utilized from the multidisciplinary scientific armamentarium. Combination algorithms, scoring systems as well as artificial intelligent models have been so far proposed for cervical screening and management. The impact of sexual lifestyle inherently possesses a key role in the prevalence of HPV-related biomarkers. This study aimed to investigate any possible influence of sexual behavior and demographic characteristics in the expression of HPV-related biomarkers in a colposcopy population from October 2016 to June 2017, and corroborated the determining role of age at fist intercourse; the older the age, the lower the probability for DNA positivity. Multivariate analysis illustrated additionally that a number of sexual partners exceeding 4.2 was crucial, with women with ≤5 partners being approximately four times less likely to harbor a positive HPV DNA test (*p* < 0.0001). Similarly, a reported partner change during the last year before HPV DNA assessment contributed to 2.5 times higher odds for DNA positivity (*p* = 0.0006). From this perspective, the further development and validation of scoring systems quantifying lifestyle factors that could reflect cervical precancer risk seems paramount.

## 1. Introduction

Cervical cancer remains the fourth most frequently diagnosed cancer and the fourth leading cause among cancer deaths, with an estimated 604,000 new cases and 342,000 deaths worldwide in 2020 [1]. In spite of robust evidence that persistent infection with high-risk human papillomavirus genotypes (hrHPV) represents a necessary but not sufficient cause of cervical cancer, other important cofactors including some sexually transmitted infections, such as HIV and Chlamydia trachomatis, smoking, higher number of lifetime childbirths and long-term use of oral contraceptives, have been linked subsequently to hrHPV acquisition and development of cervical pre-cancer and cancer [2,3,4,5].

The deployment of cytology-based cervical cancer screening strategies with early detection and treatment of precancerous lesions has significantly reduced both the incidence and mortality from cervical cancer in the developed world [1,6]. However, the current understanding is that the implementation of hrHPV testing for primary cervical screening offers significantly greater protection against invasive cancer than cytology-based screening [7,8]. The breakthrough in cervical cancer primary prevention came 15 years ago when the era of HPV prophylactic vaccines commenced. Despite Greece being among the first countries to incorporate a free of charge anti-HPV vaccination (2-valent and 4-valent vaccines) schedule in the national immunization program, the existing data on vaccination coverage are disappointing with just approximately 30%–35% of the eligible target-group cohorts having received the full regimen until recently [9].

Furthermore, in Greece, despite several initiatives, no national cervical screening program is currently in place, and most women are being opportunistically screened, predominantly by cytology and lately by HPV DNA or mRNA co-testing [10]. According to the standard practice, individuals with cytological abnormalities are subsequently referred for further assessment mainly to specialized colposcopy centers.

We conducted a study in a colposcopy clinic population of women of reproductive age, studying the expression of HPV-related biomarkers (HPV DNA and mRNA E6 and E7) in relation to dependent variables such as cytology and colposcopy, using as gold standard cervical histology (for all cases this was available—using as cut-off LGSIL+). We also sought to determine possible co-variations with demographic characteristics and lifestyle sexual factors (age of onset of sexual activity, lifetime number of sexual partners, condom use).

## 2. Materials and Methods

### 2.1. Study Population—Inclusion and Exclusion Criteria

We conducted a prospective pragmatic observational study within the framework of a multidisciplinary research protocol in cervical pathology (Ministry of Education and Religious Affairs), under the frame of the HPVGuard research project (http://HPVGuard.org, accessed on 25 June 2021, Project Number: 11ΣΥΝ_10_250, Cooperation framework, Protocol Number: ΕΥΔΕ—ΕΤAΚ 1788/1-10-2012). The study included women of reproductive age who were referred in the Larisa Primary Health Centre Colposcopy clinic for evaluation of abnormal cytology. This clinic is affiliated with the University Hospital of Larisa, covering a large proportion of referrals from Central Greece. Experienced gynecologists and board-accredited colposcopists staff the aforementioned department.

We included all women of reproductive age who signed the informed consent form harboring any grade of cytological abnormalities and/or abnormal colposcopy (LGSIL/HGSIL). We also included individuals who tested positive for HPV DNA genotyping (measured by the CLART-2 HPV test^®^ (Genomica, Madrid, Spain)) or either positive or negative mRNA E6 and E7 test as tested by APTIMA^®^ HPV Assay (Hologic, Marlborough, MA, USA) at the initial visit. There were no prerequisites regarding the anti-HPV vaccination status in terms of the eligibility to participate in the study.

We excluded individuals who were pregnant at the time of enrolment, those who had previously underwent ablative or destructive treatment of cervical precancer, and those who were previously reviewed in colposcopy for abnormal cytology.

For the purposes of the particular analysis we included individuals who were recruited in an 18-month span, from October 2016 until June 2017.

### 2.2. Study Protocol

Based on the study protocol, a detailed medical and gynecological history was obtained at the first visit in all women, covering aspects regarding age at coitarche, parity, number of lifetime sexual partners, any recent change in sexual partnership, use of condoms, and HPV immunization. In addition to epidemiological data, other confounding factors affecting HPV and cervical intraepithelial neoplasia (CIN) (e.g., smoking) were also recorded. 

Furthermore, in all participating individuals a liquid-based cytology (LBC) sample was obtained using a Rovers™ Cervex-brush just prior to the colposcopic evaluation. This was transferred in PreservCyt solution and subsequently underwent cytological and bio-molecular analysis for established HPV-related biomarkers, in particular:HPV DNA genotyping (CLART-2 HPV test^®^ (Genomica, Madrid, Spain))Detection of E6/E7 mRNA from the 14 high-risk HPV types (APTIMA^®^ HPV Assay, (Hologic, Marlborough, MA, USA)).

The cytological examination was expressed according to the Bethesda classification (TBS 2001 system) [11,12]. The HPV DNA typing using the CLART^®^ HPV2 allows the simultaneous detection of 35 different HPV genotypes (both high- and low-risk, including the subtypes included in this study) by PCR amplification of a fragment within the highly conserved L1 region of the virus [13]. The APTIMA^®^ HPV Assay is a commercially distributed kit allowing the identification of E6 and E7 mRNA of the 14 high-risk HPV types (16, 18, 31, 33, 35, 39, 45, 51, 52, 56, 58, 59, 66 and 68) utilizing a transcription mediated amplification of viral mRNA, while testing and analysis were basically performed on an automated system (Panther, Hologic) by using 1 mL from the ThinPrep sample transferred into an APTIMA Specimen Transfer tube. All specimens were analyzed for cytological and biomolecular evaluation, centrally at the University General Hospital “Attikon” of Athens.

Women with referral abnormal cytology of any grade as well as those who tested HPV DNA and/or mRNA E6 and E7 positive underwent colposcopic evaluation. Women with negative colposcopy had no biopsies taken and were referred for a repeat cytological and colposcopic assessment six months later. Women with low-grade colposcopy underwent colposcopically directed single or multiple punch biopsies. All women with high-grade colposcopy had either biopsies taken or loop excision of the transformation zone (LLETZ) of the cervix. LLETZ conization is considered the gold standard for histological diagnosis since it includes the entirety of the cervical transformation zone and was preferred over multiple punch biopsies in women who had completed childbearing [14]. The subgroup of women with low-grade colposcopy who did not have biopsies taken, after being appropriately counselled, opted for review in the colposcopy clinic in six months with repeat cytology. 

As stated, based on the study design all women underwent a colposcopic evaluation in order to document any cytologic and/or biomolecular discrepancies. All colposcopic examinations were performed by expert board-accredited colposcopists. 

In all HPV unvaccinated individuals, a strong recommendation for HPV vaccination was given irrespectively to the study’s cytological and or colposcopical findings, since this represents universal standard clinical policy of the department.

All women were informed about the scope of the study and were asked to undersign a consent form before entering the study. The study’s protocol has been approved by the Greek Central Government and subsequently received additional approval from the coordinating authority “Attikon” University Hospital Ethics Committee (Code: ΕΒΔ 623/14-5-13) [15].

### 2.3. Statistical Analysis

Statistical analysis was performed via the SAS for Windows 9.4 software platform (SAS Institute Inc., Cary, NC, USA). Descriptive values were expressed as mean ± standard deviation (SD) also reporting minimum and maximum values, as well as median and first and third quartiles (Q1, Q3) and for the categorical data using frequencies and percentages. Comparisons between groups for the qualitative parameters were made using the chi-square test and if required the Fisher exact test. We examined the relation between the number of sexual partners and the coitarche using the Spearman correlation coefficient since normality could not be ensured for these variables. We evaluated if these two characteristics can be used as single predictors of HPV DNA and mRNA positivity using the receiver operating characteristics (ROC) area under curve (AUC). The significance level for all statistical tests was set to *p* < 0.05 and all tests were two sided.

The performances of cytology and colposcopy, as well as HPV DNA and mRNA detection in respect to the histological results, were set as primary outcomes of the study, and were assessed by counting the true positive and negative cases as well as the false positive and negative cases for various cytological and colposcopical cut-offs as well as for various histological thresholds. Subsequently, we calculated the sensitivity, specificity, positive and negative predictive value (PPV and NPV, respectively), and false positive and false negative rates (FPR and FNR, respectively). Moreover, as single number indicators of the overall performance we calculated the overall accuracy and the diagnostic odds ratios (DOR).

## 3. Results

### 3.1. Demographic Data

In total 336 women participated in the study; both their demographic and medical baseline characteristics are depicted in Table 1. The median age of the participating women was 28 years, (first and third quartiles (Q1–Q3: 24–34 years)) with minimum age 18 years and maximum 48 years. The parity rate was relatively low (0.16%), with only 8 women (representing 2.4%) being para 2 or 3.

At enrollment, about one in three (36.9%) were current smokers and similarly 34.5% were vaccinated, the vast majority (81%) with quadrivalent Gardasil. On average, the number of lifetime sexual partners was 6; 33.9% of the women had ≤3 lifetime partners and a similar percentage of 36.3% had more than 5. For the whole population, the average percentage of condom use during intercourse was 29.8%; 150 women were never users (44.6%) and 14 (4.2%) were consistent users (100% rate of condom utilization).

In terms of medical conditions, this special colposcopy clinic population illustrated 53.6% aberrant cytology rates, while an even higher percentage (62.5%) harbored abnormal colposcopic findings. HPV DNA was found positive in 152 (45.24%) of the participating women, and of these 60 (39.5% of the HPV positive population) had co-infection with multiple HPV genotypes (36 had double, 12 triple, 6 quadruple, 2 five-fold and 2 six-fold co-infection). In addition, 92 women (27.4%) were found to be HPV mRNA positive while no further mRNA typing was performed. Finally, history of warts (active or treated) was reported in 18 women (5.4%).

Regarding the expression of HPV’s genotypes in our population, the results of the HPV DNA assay detected a distribution of particular HPV subtypes that concurs with reported geographical data [16,17]. Almost 20% of the women tested positive for hrHPV 51 (30/152; 19.7%), while hrHPV 66 was prevalent to a similar extent, being expressed in 28 out of 152 (18.4%) of the HPV positive individuals. Furthermore, hrHPV31 was detected in 24 women (15.8%), while an identical percentage of study individuals featured hrHPV68. HPV 16 was found in 18 women (11.8%); only 8 individuals harbored HPV 18 (11.8%) and, finally, 10 out of 152 women tested positive for HPV45 (6.6%) (Figure 1).

### 3.2. Factors Related to HPV DNA

#### 3.2.1. Univariate Analysis

In the sequel, we investigated the relation of a positive vs. negative HPV DNA result in regard to all other recorded patient characteristics, in terms of odds ratios (ORs). The results are presented in Table 2; in particular, the second column reports the contingency tale between HPV DNA negative or positive result and the values of each studied parameter. For instance, (as anticipated) a negative HPV DNA test was strongly (*p* < 0.0001) related with negative cytology, (See Table 2) whereas 118 from the 184 (i.e., 64.13%) negative HPV DNA results were also cytology-negative, and the remaining 66 (35.9%) were HPV DNA negative and had an abnormal Papanicolaou test; similarly, 75% of the HPV DNA positive women also had an abnormal cytology (OR: 5.4, 95%CI: 3.3–8.6). Moreover, abnormal colposcopic impression was also strongly related with a positive HPV DNA result (OR: 72.8, 95% CI: 25.8–205.8), as was also the case for HPV mRNA (OR: 61.9, 95% CI: 21.8–175.4). Furthermore, 97.4% of individuals with HPV DNA positivity presented with abnormal colposcopic findings. The concordance between HPV assays was substantial, with 97.8% of HPV DNA negative women having a negative HPV mRNA test. 

In our study, the number of sexual partners gained importance for HPV DNA positivity when it exceeded the threshold of three partners (per lifetime)—actually for this cut-off the OR was 1.9 (95% CI: 1.2–3.0); in other words, women with more than three partners per lifetime had two times higher odds for a positive HPV DNA test. Notably, the odds were increasing with increasing numbers of sexual partners, for instance for five partners OR reached 3.8 and for 10 OR reached 4.2. In terms of sexual behavior, a recent (within the last 12 months) change of partner was also predictive of a positive HPV DNA outcome (OR: 2.1, 95% CI: 1.2–3.8, *p* = 0.0108). Actually, if we use the number of lifetime sex partners as a predictor for HPV DNA positivity then the AUC of the ROC curve was 65.39% (95% CI: 59.39–71.39%) indicative that is a rather good predictor. Moreover, for the age of coitarche yielded an AUC of 44.88% (95% CI: 38.80–50.96%).

Interestingly, vaccination status, history of warts and condom use did not yield statistical significance in our study (*p* > 0.05 for all parameters). 

Oddly in our patients smoking appeared to a play a “protective” role (OR: 0.5, *p* = 0.0014), as 55% of HPV DNA negative women were smokers, in contrast to the higher percentage (72%) of HPV DNA positive women who were non-smokers, and this could be perhaps attributed to the relatively small study sample [18,19] (See Table 2). 

#### 3.2.2. Multivariate Analysis

Demographic characteristics which were found to be important in the univariate analysis with a *p*-value < 0.2 were entered in a multivariate analysis model in order to control for confounders. A diagram illustrating the OR for the parameters being found important using as a reference point a positive HPV DNA test is depicted in Figure 2. 

Interestingly the age of first sexual intercourse was found to be statistically important (OR: 0.8, 95% CI: 0.7–0.9, *p* = 0.0032), indicative that the higher this age the lower the probability for DNA positivity. 

Among the various cut-offs for the number of sex partners, the multivariate analysis demonstrated that a cut-off of five sexual partners was crucial; specifically, women with ≤5 partners were about four times less likely to harbor a positive DNA test (OR: 0.24, 9% CI: 0.14–0.40, *p* < 0.0001). Furthermore, a partner change during the last year before HPV DNA assessment conferred 2.5 times higher odds for DNA positivity (OR: 0.4, 95% CI: 0.2–0.8, *p* = 0.0093). Finally, both smoking (OR: 2.5, 95% CI: 1.5–4.3, *p* = 0.0006) and HPV vaccination (OR: 1.9, 95% CI: 1.1–3.2, *p* = 0.0283) had a positive impact for a negative HPV DNA outcome. 

Characteristic plots depicting graphically the role of age at first sexual intercourse and number of sexual partners vs. smoking and vaccination are depicted in Figure 3. Note that the Spearman correlation coefficient between the age at first sexual intercourse and number of sexual partners was r_s_ = −0.05 (*p* = 0.3969), indicative that these two characteristics were not related.

### 3.3. Factors Related to HPV mRNA Positivity

#### 3.3.1. Univariate Analysis

In a similar manner, as for HPV DNA, we examined the role of all the studied variables in relation to HPV mRNA positivity (see results in Table 2). Cytology results, colposcopy and HPV DNA outcomes did correlate positively with the HPV mRNA outcome, with particularly higher odds for cytology (Papanicolaou test) (OR: 14.0, 95% CI: 6.7%–29.1%, *p* < 0.0001) compared to 5.4 for HPV DNA, a finding indicative that HPV mRNA might represent a better predictor of underlying cervical pathology. All mRNA positive cases also harbored colposcopical abnormalities (therefore, ORs could not be calculated). Moreover, despite the finding that positive HPV DNA and mRNA results were strongly interrelated, the study data analysis did not detect a role of multiple infections in HPV mRNA expression rates.

In terms of the role of the number of previous sexual partners, data analysis illustrated that a history of more than one partner might affect mRNA expression (see Table 3; OR: 7.3, 95% CI: 1.7–30.1, *p* = 0.0019) and similarly there is a trend that the odds are increasing with rising number of sexual partners. The number of lifetime sex partners was also tested as a predictor for HPV mRNA positivity; the AUC of the ROC curve was 62.88% (95% CI: 56.11%–69.66%), indicative that it is a rather good predictor. Moreover, the age of coitarche yielded an AUC of 46.08% (95% CI: 39.20%–52.97%). A reverse role of smoking was also documented regarding mRNA positivity rates (OR: 0.5, *p* = 0.0116).

Regarding vaccination status, vaccinated women in our study had lower odds for a positive mRNA expression; actually, among mRNA positive women 23.9% were vaccinated, whereas among mRNA negative women 38.5% were vaccinated (OR: 2). Finally, vaccine type did not affect HPV mRNA expression rates.

#### 3.3.2. Multivariate Analysis

Similar to DNA positivity, we applied multivariate analysis for addressing any possible confounding factors influencing mRNA positivity. The factors entered in the multivariate models were those with a *p*-value < 0.1 in the univariate analysis (see Table 2). As was the case for HPV DNA, the number of partners was the most determining factor affecting mRNA positivity (OR: 1.1, 95% CI: 1.1–1.2, *p* < 0.0001), while as observed for DNA positivity, non-vaccinated women and oddly non-smokers illustrated higher rates of mRNA positivity (OR: 2.1, 95% CI: 1.2–3.8, *p* = 0.0135, and OR: 2.3, 95% CI: 1.2–4.2, *p* = 0.0099, respectively).

### 3.4. Performance in Comparison to Histological Results

Additionally, we calculated various performance indicators for the role of cytology, colposcopy, HPV DNA and HPV mRNA as tests which affect the corresponding cervical status (as reflected in the histological outcome)—note that per protocol all women which simultaneously presented with negative cytology, HPV DNA and colposcopy were considered as being histologically negative. These results are summarized in Table 3 and Table 4.

Clearly, regarding cytological and histological cut-offs of ≥LSIL, sensitivity of cytology reached 72% while specificity was above 87% (see Table 3).

Similar to cytology, a colposcopical and histological threshold of ≥LSIL provided rather balanced results (DOR:10.9, Table 4) and when considering the HSIL + threshold, despite the excellent specificity (100%), the sensitivity reached 75%.

## 4. Discussion

The implementation of HPV-related biomarkers to augment early detection and treatment of pre-invasive cervical lesions has constantly triggered the scientific efforts of our group [20,21,22,23,24,25,26,27,28,29]. Extensive research has been published during the last decade adopting different approaches under the concept and umbrella of everyday clinical practice through pragmatic and feasibility studies [6,30,31,32,33]. Such approaches benefit from appreciating acceptance and usefulness in periods of financial crisis as well as during periods when financial resources are re-allocated (for instance during the COVID-19 pandemic), especially in identifying individuals at higher risk to be triaged for immediate colposcopy referral and/or treatment [28]. 

A plethora of HPV-related biomarkers have been examined so far by our group, investigating either the expression profile of those markers and possible applications in surveillance, management and treatment of cervical precancer [6,34,35,36,37,38,39,40]. Other groups have studied novel diagnostic approaches such as biospectroscopy, showing promising results and interesting potential [41,42,43].

The low overall prevalence of HG disease in this cohort predictably had an impact on our findings; the main study’s results are in line with the existing epidemiological literature [44,45]. In a recent publication, Sanjose et al. highlighted three determinants of the HPV prevalence by age group: a) age at coitarche, b) number of sexual partners and c) concomitant sexual partners and screening history. The two former factors have been also corroborated in our study, where higher age of first sexual intercourse was related with lower probability for DNA positivity; furthermore, a cut-off of five sexual partners was crucial, with women with ≤5 partners being approximately four times less likely to harbor a positive DNA test (*p* < 0.0001) [46].

In our study a partner change during the last year before HPV DNA assessment contributed to 2.5 times higher odds for DNA positivity (*p* = 0.0006), a finding also consistent with the literature [44]. In the older 12-month prospective study by Giuliano et al., the risk of incident HPV infection (acquired after baseline) was strongly influenced by the number of new male sex partners, illustrating a comparable 2.39 relative hazard [47].

Particularly in recent times, the clinician should carefully balance between a too intrusive or disengaged approach during history taking in the colposcopy clinic; the presence of a midwife is helpful in this direction. After reassuring the patient on privacy concerns and that all data will be anonymized, women should be encouraged to provide a scheduled history covering basic aspects of reproductive and sexual demographic data. Epidemiological co-factors which augment hrHPVs in the development of severe precancerous lesions possess an established all-important role in cervical carcinogenesis which should not be overlooked or even dismissed whatsoever, in light of the elucidated role of hrHPV-mediated molecular mechanisms [44]. Another advantage of epidemiological parameters is linked to their inherent longitudinal stability, in contrast with HPV-related biomarkers which could fluctuate over time [46,48].

From this perspective, the further development and validation of scoring systems quantifying lifestyle factors that could impact cervical precancer risk seems paramount [49]. The recent publications of our group contributed in this direction [30,50]. The seminal paper of Paraskevaidis et al. identified four key lifestyle risk factors (age at onset of sexual intercourse, number of sexual partners, smoking and frequency of condom use) [51]. These lifestyle factors were stratified in a three-tier risk band (low-, middle- or high-risk) and coupled with demographic data (age, education level), referral cytology and HPV biomarkers (mostly mRNA) to augment clinical decisions regarding whether treatment (usually large loop excision of the transformation zone (LLETZ)) is warranted [36]. As illustrated in the paper of Paraskevaidis et al., these lifestyle scoring systems can be coupled with combinations of molecular markers to yield parametric risk assessment tools with predictive abilities. An integration of lifestyle scoring systems together with artificial intelligence tools for predicting underlying cervical histology represents an ambitious, yet feasible approach [34,35,52].

## 5. Conclusions

Research in emerging HPV-related biomarkers should be coupled with the development, integration and validation of lifestyle scoring systems. These systems quantify lifestyle factors that could impact cervical precancer risk and together with artificial intelligence tools accurately predict underlying cervical pathology.

## Figures and Tables

**Figure 1 biology-10-00713-f001:**
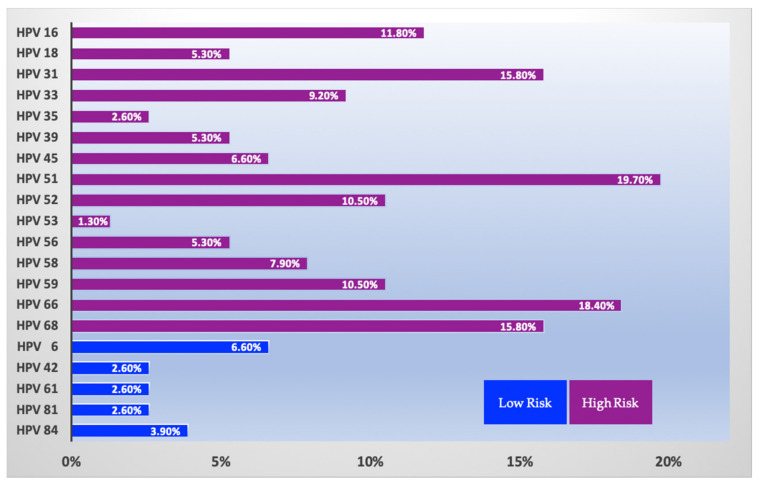
Distribution of HPV subtypes in HPV DNA positive women of the study population.

**Figure 2 biology-10-00713-f002:**
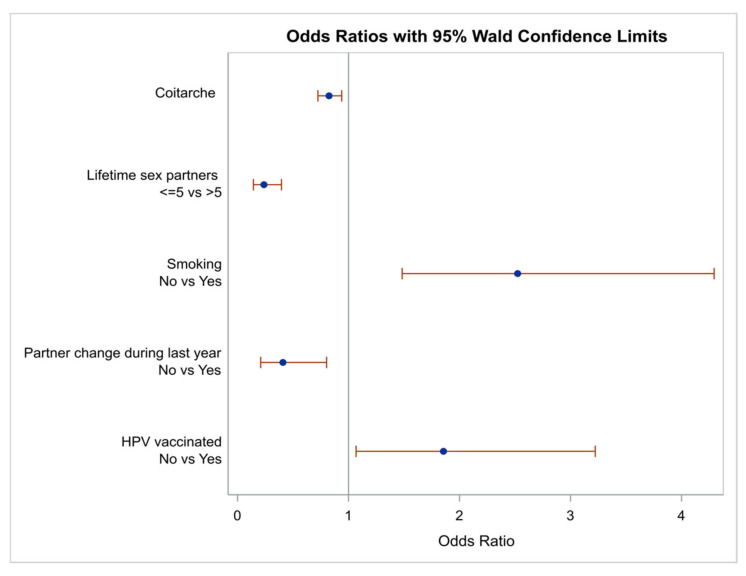
Odds ratios and 95% confidence limits of the demographic characteristics which were found important for DNA positivity at multivariate analysis. Note that ORs are the reciprocal of the ORs for positive test results.

**Figure 3 biology-10-00713-f003:**
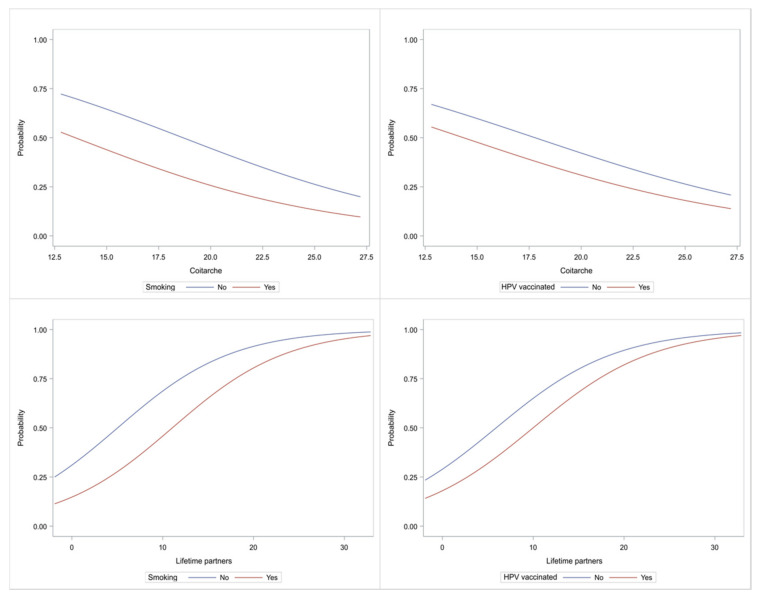
Characteristic curves showing the probability for DNA positivity. Top row: age of first sexual intercourse; Bottom row: number of sex partners; Left column: smoking; Right column: vaccination. Vertical axis shows the probability for DNA positivity.

**Table 1 biology-10-00713-t001:** Baseline characteristics of the study population.

Characteristic	Value
Total population in the study (N)	336
Age (mean ± SD, minimum, maximum)	28.8 ± 6.3, 18–48
Smoking (N, %)	124 (36.9%)
Parities (mean ± SD, minimum, maximum)	0.16 ± 0.57, 0–3
HPV vaccination (N,%)	116 (34.52%)
Cervarix	22 (18.97%)
Gardasil	94 (81.03%)
Lifetime number of sexual partners (mean ± SD, minimum, maximum)	6.0 ± 4.9, 1–30
Percentage of condom use (mean ± SD, minimum, maximum)	29.8% ± 32.1%, 0–100%
Change of sexual partner during the last year (N, %)	56 (16.7%)
Test Papanicolaou results (N, %)	
NILM	156 (46.4%)
LSIL (includes HPV)	132 (39.3%)
ASCUS	40 (11.9%)
HSIL	8 (2.4%)
Colposcopy results (N, %)	
Negative colposcopy—Adequate/Normal colposcopic findings	126 (37.5%)
LSIL (includes HPV)	204 (60.7%)
HSIL	6 (1.8%)
Histology results (N, %)	
Unavailable (No histological specimen obtained) *	116 (34.5%)
Normal (Negative histology)	42 (12.5%)
LSIL (includes HPV)	170 (50.6%)
HSIL	8 (2.4%)
Ca	0
HPV DNA result (N, %)	
Negative	184 (54.8%)
Positive	152 (45.2%)
HPV mRNA result (N, %)	
Negative	244 (72.6%)
Positive	92 (27.4%)

* Histological examination was not performed; these cases presented with normal cytology and colposcopy and were HPV DNA negative.

**Table 2 biology-10-00713-t002:** Statistical comparisons of HPV DNA positive vs. negative result with all other recorded variables. Bold entries indicate statistical significance.

	HPV DNA Positivity vs. Parameter Levels (N/%), HPV DNA Value in Rows	*p*	Odds Ratio (95%CI)	HPV mRNA Positivity vs. Parameter Levels (N/%), HPV mRNA Value in Rows	*p*	Odds Ratio (95%CI)
LBC ^1^	Negative: NILM (118/64.1%), Positive: NILM (38/25%)	**<0.0001**	NA	Negative: NILM (147/60.3%), Positive: NILM (9/9.8%)	**<0.0001**	NA
	Negative: ASCUS (30/16.3%), Positive: ASCUS (10/6.6%)			Negative: ASCUS (34/13.9%), Positive: ASCUS (6/6.5%)		
	Negative: LSIL (36/19.6%), Positive: LSIL (96/63.2%)			Negative: LSIL (63/25.8%), Positive: LSIL (69/75%%)		
	Negative: HSIL (0/0%), Positive: HSIL (8/5.3%)			Negative: HSIL (0/0%), Positive: HSIL (8/8.7%)		
Abnormal LBC	Negative: 66/35.9% Positive: 114/75%	**<0.0001**	5.4 (3.3–8.6)	Negative: 97/39.8% Positive: 83/90.2%	**<0.0001**	14.0 (6.7–29.1)
LBC threshold	Negative: LSIL+ (36/19.6%) Positive: LSIL+ (104/68.4%),	**<0.0001**	8.9 (5.4–14.7)			
Colposcopic Severity Findings	Negative: Negative (122/66.3%), LSIL (62/33.6%), HSIL (0/0%) Positive: Negative (4/2.6%), LSIL (142/93.4%), HSIL (6/3.9%)	**<0.0001**	NA	Negative: Negative (126/51.6%), LSIL (118/48.4%), HSIL (0/0%), Positive: Negative (0/0%), LSIL (86/93.5%), HSIL (6/6.5%)	**<0.0001**	NA
Abnormal colposcopy	Negative: 62/33.7% Positive: 148/97.4%	**<0.0001**	72.8 (25.8–205.8)	Negative: 118/48.3% Positive: 92/100%	**<0.0001**	
HPV mRNA positive	Negative: 4/2.2% Positive: 88/57.9%	**<0.0001**	61.9 (21.8–175.4)	See left column		
Lifetime Partners > 3	Negative: 110/59.8% Positive: 112/73.7%	**0.0074**	1.9 (1.2–3.0)	Negative: 152/62.3% Positive: 70/76.1%	**0.0173**	1.9 (1.3–3.3)
Partner change during last year	Negative: 22/12% Positive: 34/22.4%	**0.0108**	2.1 (1.2–3.8)	Negative: 40/16.4% Positive: 16/17.4%	**0.0479**	1.1 (0.6–2.0)
Condom use	Negative: 74/40.2% Positive: 72/47.4%	0.1881	1.3 (0.9–2.1)	Negative: 104/42.6% Positive: 42/45.7%	0.2495	1.1 (0.7–1.8)
Smoking	Negative: 82/44.6% Positive: 42/27.6%	**0.0014**	0.5 (0.3–0.8)	Negative: 100/41% Positive: 24/26.1%	6.367	0.5082 (0.3–0.7)
HPV Vaccinated	Negative: 70/38% Positive: 46/30.3%	0.1354	0.7 (0.4–1.1)	Negative: 94/38.5% Positive: 22/23.9%	6.31	0.5 (0.3–0.7)

^1^ LBC: Liquid-based cytology. Bold letters indicate statistical significance

**Table 3 biology-10-00713-t003:** Performance indicators of Papanicolaou test using as gold standard the histological outcome.

Cytological and Histological Cut Off	≥LSIL	≥HSIL
Sensitivity	71.8%	100.0%
Specificity	87.5%	87.8%
PPV	60.7%	16.7%
NPV	92.1%	100.0%
FPR	12.5%	12.2%
FNR	28.2%	0.0%
OA	84.2%	88.1%
Diagnostic Odds Ratio	17.93	NA

**Table 4 biology-10-00713-t004:** Performance indicators of colposcopy using as gold standard the histological outcome.

Colposcopical and Histological Cut Off	≥LSIL	≥HSIL
Sensitivity	69.0%	75.0%
Specificity	83.0%	100.0%
PPV	52.1%	100.0%
NPV	90.9%	99.4%
FPR	17.0%	0.0%
FNR	31.0%	25.0%
OA	80.1%	99.4%
Diagnostic Odds Ratio	10.9	NA

## Data Availability

Data are available from the corresponding author upon a reasonable request.

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
