# Peer review of "The Influence of Sexual Behavior and Demographic Characteristics in the Expression of HPV-Related Biomarkers in a Colposcopy Population of Reproductive Age Greek Women"

_biology, 2021, doi:10.3390/biology10080713_

Round 1

Reviewer 1 Report

Valasoulis et al. have explored HPV DNA and HPV mRNA expression in cervical samples regarding sexual behavior and demographic characteristics from 336 women referred to colposcopy. Women with abnormal cytology, women with more than five lifetime sexual partners, women with a new partner during last year and women reporting smoking have higher positivity rate in HPV DNA and HPV mRNA test than other women.

Valasoulis et al. also presented performance indicators of cervical cytology and HPV-test in different combinations and different cut-offs using histology outcome as a gold standard.

Comments

There are no histology results presented. How many of the 336 women had CIN2, CIN3 or cervical cancer? Why use both HPV DNA and HPV mRNA test? Could the HPV mRNA test be used as a triage test of HPV DNA positive? Is it possible to use the HPV mRNA test as a stand alone test without HPV DNA testing? Co-testing with both cervical cytology and HPV DNA test is not recommended. Using HPV DNA test in primary screening, is HPV mRNA or cytology a better triage test? What cut-off of cytology is recommended in triage of HPV DNA positive? Which algorithm would you suggest maximizing benefits and reducing harms of screening?

It is known that number of sexual partners, change of partner and smoking are risk factors for acquiring HPV-infections. But if put all factors in a multivariate analysis with histologically confirmed CIN3+ as endpoint, will all the risk factors for CIN3+ remain, or is persistent HPV-infection the only reason for CIN3+?

Page 1, abstract, include the number of women (336), number of abnormal cytology 53.6% (180/336), the HPV DNA result 54.8% (184/336), HPV mRNA result 27.4% (92/336), the detection rate of histologically confirmed CIN3+ xx% (yy/336).

Page 4, Table 1, use only one decimal in percentages (34.52% => 34.5%)

Page 5, Table 1, please use the Bethesda classification of cervical cytology (LBC). HPV is not a Bethesda diagnosis (Normal, ASC-US, LSIL, ASC-H, HSIL)

Page 6, Figure 1, why include Low Risk HPV-types? lrHPV-types are not a risk factor for CIN3+. Use only one decimal in percentages (11.80% => 11.8%)

Page 7, Table 2, use only one decimal in percentages (16.304% => 16.3%)

Page 7, Table 2, LBC, it gives no meaning to present distribution of cytology diagnoses in HPV negative and HPV positive women. If you want to present cytology results, make a table with Normal cytology (HPV +/-), ASC-US (HPV +/-), LSIL, ASC-H (HPV +/-) and HSIL (HPV +/-).

Page 7, Table 2, LBC threshold, according to Bethesda classification LSIL is more severe than ASC-US. There is no meaning to present ASC-US+ versus LSIL-. One cut-off is ASC-US+ versus Normal, or LSIL+ versus ASC-US-, or ASC-H+ versus LSIL-.

Page 7, Table 2, Lifetime Partners, keep >5, remove >1, >2, >3, >10. The table is messy enough without a lot of cut-offs regarding number of partners.

Page 7, Table 2, remove “HPV Vaccine type” and “History of Warts”

Page 9-10, remove Table 3, it is too much overlap between HPV DNA and HPV mRNA results.

Page 11, Table 4, “HPV in histology” or “LSIL+ in histology” gives no meaning. The gold standard is high-grade histology (CIN2+ or CIN3+). A CIN1 diagnosis does not represent a significant risk factor for CIN3 above the risk attributed to its molecular cause, genotype-specific HPV infection. CIN1 should not be a target of screening and women with CIN1 should not be treated (Castle 2011).

Page 11, Table 4, “WLN and HPV is negative”, HPV is not a diagnosis of cytology in the Bethesda system. Using the term “negative HPV” can be inter-pretended as a negative HPV DNA test. Women with a negative HPV DNA test should not be referred to colposcopy.

In my opinion both Table 4 and Table 5 are confusing and misleading. It is better to make 2 x 2 tables with cytology result at different cut-offs versus histology results (CIN2+), see Sørbye 2011 (Table 2 and 3).

Page 12, discussion, “Despite the low overall prevalence of HG disease in this cohort”. Have you presented the prevalence of CIN2+ or CIN3+ at all? I have not found the results, but in Table 1 there is reported only six women with HSIL colposcopy results. Is this the histology results?

Page 13, conclusions, “Research in emerging HPV-related biomarkers should be coupled with the development, integration and validation of lifestyle scoring systems”

Did you try to make a scoring system of the sexual and demographic characteristics in your material? How many of women with CIN3+ were found using lifstyle score and how many were found base on abnormal cytology and a positive HPV-test?

Page 13, conclusions, “These systems quantify lifestyle factors that could impact cervical precancer risk and together with artificial intelligence tools accurately predict underlying cervical pathology”

Can you conclude regarding artificial intelligence tools when you have not used any AI tools in the manuscript?

I think primary cervical cancer prevention using HPV-vaccination, and secondary prevention using HPV-screening and cytology and/or molecular biomarkers in triage is more reliable than lifestyle factors scoring systems.

References

Castle PE, Gage JC, Wheeler CM, Schiffman M. The clinical meaning of a cervical intraepithelial neoplasia grade 1 biopsy. Obstet Gynecol. 2011 Dec;118(6):1222-1229. doi: 10.1097/AOG.0b013e318237caf4.

https://pubmed.ncbi.nlm.nih.gov/22105250/

Sørbye SW, Arbyn M, Fismen S, Gutteberg TJ, Mortensen ES. HPV E6/E7 mRNA testing is more specific than cytology in post-colposcopy follow-up of women with negative cervical biopsy. PLoS One. 2011;6(10):e26022. doi: 10.1371/journal.pone.0026022.

https://pubmed.ncbi.nlm.nih.gov/21998748/

Author Response

Reviewer 1

Comments and Suggestions for Authors

Valasoulis et al. have explored HPV DNA and HPV mRNA expression in cervical samples regarding sexual behavior and demographic characteristics from 336 women referred to colposcopy. Women with abnormal cytology, women with more than five lifetime sexual partners, women with a new partner during last year and women reporting smoking have higher positivity rate in HPV DNA and HPV mRNA test than other women.

Valasoulis et al. also presented performance indicators of cervical cytology and HPV-test in different combinations and different cut-offs using histology outcome as a gold standard.

Comments

Comment 1: There are no histology results presented. How many of the 336 women had CIN2, CIN3 or cervical cancer?

Authors’ actions: Thank you for your comment. In our study population 170 individuals (170/336, 50.6%) had histologically confirmed CIN1 and 8 women (8/336, 2.4%) histologically confirmed CIN2+. This data is clearly presented in Table 1.

Comment 2: Why use both HPV DNA and HPV mRNA test?

Authors’ actions: Thank you for your comment. As stated in the Materials and Methods Section this study was conducted within the framework of a Greek multidisciplinary research protocol in cervical pathology; the implementation of both assays was based on the study protocol.

Comment 3: Could the HPV mRNA test be used as a triage test of HPV DNA positive?

Authors’ actions: We appreciate your comment; recent research from the NTCC2 group has addressed this question concluding that other HPV-related biomarkers might be more suitable for this purpose (Giorgi-Rossi P et al, JNCI 2021 doi: 10.1093/jnci/djaa105).

Comment 4: Is it possible to use the HPV mRNA test as a standalone test without HPV DNA testing?

Authors’ actions: Thank you for your comment. Several publications show a potential for primary cervical screening implementing mRNA HPV assays, of special interest is the recent work by Weston G et al, BMJ Open 2020 doi:10.1136/bmjopen-2019-031303

Comment 5: Co-testing with both cervical cytology and HPV DNA test is not recommended. Using HPV DNA test in primary screening, is HPV mRNA or cytology a better triage test?

Authors’ actions: Thank you for your remark. At the time the study was designed there was insufficient evidence to recommend HPV DNA testing as a primary screening method. Currently, primary HPV DNA testing with reflex cytology represents the most common HPV-based strategy in Western countries, as it provides higher sensitivity and acceptable colposcopy referral rates compared with cytology (Ogilvie GS et al, JAMA 2018 doi:10.1001/jama.2018.7464/Leinonen MK et al, BMJ 2012 doi: 10.1136/bmj.e7789).

Comment 6: What cut-off of cytology is recommended in triage of HPV DNA positive? Which algorithm would you suggest maximizing benefits and reducing harms of screening?

Authors’ actions: Thank you for your comment. In this particular study we implemented LSIL and HSIL. Besides, cost effective algorithms depend on the infrastructures and settings and the availability of competent colposcopists. However, please consider that these interventions were part of a scientific protocol.

Comment 7: It is known that number of sexual partners, change of partner and smoking are risk factors for acquiring HPV-infections. But if put all factors in a multivariate analysis with histologically confirmed CIN3+ as endpoint, will all the risk factors for CIN3+ remain, or is persistent HPV-infection the only reason for CIN3+?

Authors’ actions: Thank you indeed for your comment. Given the low prevalence of high grade histology in our study population, we considered that a multivariate analysis with CIN3 endpoint would be statistically problematic; therefore, we used persistent HPV DNA positivity as an endpoint.

Comment 8: Page 1, abstract, include the number of women (336), number of abnormal cytology 53.6% (180/336), the HPV DNA result 54.8% (184/336), HPV mRNA result 27.4% (92/336), the detection rate of histologically confirmed CIN3+ xx% (yy/336).

Authors’ actions: Thank you for your comment. We have addressed these points in the revised manuscript. Please note that within the study population 8 CIN2+ cases were confirmed by histology (8/336, 2.4%) – please also see Comment 1.

Comment 9: Page 4, Table 1, use only one decimal in percentages (34.52% => 34.5%)

Authors’ actions: Thank you. We have addressed these points in the revised manuscript.

Comment 10: Page 5, Table 1, please use the Bethesda classification of cervical cytology (LBC). HPV is not a Bethesda diagnosis (Normal, ASC-US, LSIL, ASC-H, HSIL)

Authors’ actions: Thank you for your recommendation. We have now reclassified all “HPV” cases in the “LSIL” category.

Comment 11: Page 6, Figure 1, why include Low Risk HPV-types? lrHPV-types are not a risk factor for CIN3+. Use only one decimal in percentages (11.80% => 11.8%)

Authors’ actions: Thank you for your comment. The validated HPV DNA assay that has been implemented additionally assessed the expression of Low Risk genotypes; we present these data since these genotypes might contribute to the LG histology pool. Furthermore, future studies might potentially gain some insight from the presentation of these data.

Comment 12: Page 7, Table 2, use only one decimal in percentages (16.304% => 16.3%)

Authors’ actions: Thank you for your point. Table 2 was modified as suggested, moreover the two last column indicating 95% CI were integrated to the column with the OR. In addition, we removed redundant decimal points from Table 1.

Comment 13: Page 7, Table 2, LBC, it gives no meaning to present distribution of cytology diagnoses in HPV negative and HPV positive women. If you want to present cytology results, make a table with Normal cytology (HPV +/-), ASC-US (HPV +/-), LSIL, ASC-H (HPV +/-) and HSIL (HPV +/-).

Authors’ actions: Thank you for your remark. Table 2 was modified as suggested; in addition, the HPV group from Table 1 was integrated with LSIL both for cytology and colposcopy.

Comment 14: Page 7, Table 2, LBC threshold, according to Bethesda classification LSIL is more severe than ASC-US. There is no meaning to present ASC-US+ versus LSIL-. One cut-off is ASC-US+ versus Normal, or LSIL+ versus ASC-US-, or ASC-H+ versus LSIL-.

Authors’ actions: Thank you for your advice. Therefore, in the revised form only LSIL+ was benchmarked, the corresponding new OR and p-values are being reported.

Comment 15: Page 7, Table 2, Lifetime Partners, keep >5, remove >1, >2, >3, >10. The table is messy enough without a lot of cut-offs regarding number of partners.

Authors’ actions: Thank you for your remark. In the revised manuscript only the cut-off of 3 partners is now reported, since this represents the threshold of life-time partners with a “statistically” significant role. The detailed breakdown analysis of this parameter was performed to illustrate that even a marginal change of one additional partner yielded statistically significant results.

Comment 16: Page 7, Table 2, remove “HPV Vaccine type” and “History of Warts”

Authors’ actions: Thank you for your remark. Table 2 was changed as proposed. “HPV Vaccine type” and “History of Warts” have been removed.

Comment 17: Page 9-10, remove Table 3, it is too much overlap between HPV DNA and HPV mRNA results.

Authors’ actions: Thank you for your comment. In the revised manuscript Table 3 was integrated into Table 2 to highlight possible differences between HPV DNA and mRNA expression profile in the various comparison groups. Of note, Table 2 is not anymore cluttered. We hope that you also consider the integrated Table being more informative.

Comment 18: Page 11, Table 4, “HPV in histology” or “LSIL+ in histology” gives no meaning. The gold standard is high-grade histology (CIN2+ or CIN3+). A CIN1 diagnosis does not represent a significant risk factor for CIN3 above the risk attributed to its molecular cause, genotype-specific HPV infection. CIN1 should not be a target of screening and women with CIN1 should not be treated (Castle 2011).

Authors’ actions: Thank you for comment. Indeed, CIN1 is not the target of screening; persistent histological CIN1 is being treated only occasionally under special circumstances. However, the study protocol did not include assessment of the vaginal microbiome and proteomics as well as metabolomics to gain some insight on the potential for deterioration and the outcome of these LG lesions.

Comment 19: Page 11, Table 4, “WLN and HPV is negative”, HPV is not a diagnosis of cytology in the Bethesda system. Using the term “negative HPV” can be inter-pretended as a negative HPV DNA test. Women with a negative HPV DNA test should not be referred to colposcopy.

Authors’ actions: Thank you for comment. Table 4 has now been revised (re-numbered Table 3). We use cut-offs of ³LSIL & ³HSIL to estimate the performance of cytology and colposcopy (re-numbered Table 3 & Table 4).

Comment 20: In my opinion both Table 4 and Table 5 are confusing and misleading. It is better to make 2 x 2 tables with cytology result at different cut-offs versus histology results (CIN2+), see Sørbye 2011 (Table 2 and 3).

Authors’ actions: Thank you for your comment. We have revised, as requested, the re-numbered Tables 3 & 4. However, we do not believe that including two additional Tables as presented in the publication of Sorbye et al. could enhance the interpretation of our results.

Comment 21: Page 12, discussion, “Despite the low overall prevalence of HG disease in this cohort”. Have you presented the prevalence of CIN2+ or CIN3+ at all? I have not found the results, but in Table 1 there is reported only six women with HSIL colposcopy results. Is this the histology results?

Authors’ actions: Thank you for your comment. This point has already been addressed in our rebuttal in Comment 1 & Comment 8.

Comment 22: Page 13, conclusions, “Research in emerging HPV-related biomarkers should be coupled with the development, integration and validation of lifestyle scoring systems”. Did you try to make a scoring system of the sexual and demographic characteristics in your material? How many of women with CIN3+ were found using lifestyle score and how many were found based on abnormal cytology and a positive HPV-test?

Authors’ actions: Thank you for your comment. The evaluation of the performance of a lifestyle scoring system in identifying high grade (HG) cervical disease was outside the scopes of this particular study, especially given the low overall prevalence of HG histology in this cohort. However, our group has gained some insight from the implementation of lifestyle scoring system as illustrated in our previous work (Refs 50, 51, 52 and 53).

Comment 23: Page 13, conclusions, “These systems quantify lifestyle factors that could impact cervical precancer risk and together with artificial intelligence tools accurately predict underlying cervical pathology”. Can you conclude regarding artificial intelligence tools when you have not used any AI tools in the manuscript?

Authors’ actions: Thank you for your remark. Indeed, we haven’t implemented Artificial Intelligence tools in this study, however previous experience with clinical decision support system for patient-specific prediction algorithms in cervical pathology that our group has previously developed and tested was most promising (Refs 50 and 53).

Comment 24: I think primary cervical cancer prevention using HPV-vaccination, and secondary prevention using HPV-screening and cytology and/or molecular biomarkers in triage is more reliable than lifestyle factors scoring systems.

Authors’ actions: Thank you for your comment. Indeed, primary and secondary cervical prevention are the principle tools against cervical cancer and are therefore integrated in the 2020 World Health Organization’s Global Strategy for this disease’s elimination. However, we do consider that adoption of “one size fits all” strategies is not without shortcomings and that scoring systems represent a valuable tool in personalized, precision medicine. In our understanding, even the popular 2020 American Society for Colposcopy and Cervical Pathology (ASCCP) app itself essentially represents a scoring system.

References

Castle PE, Gage JC, Wheeler CM, Schiffman M. The clinical meaning of a cervical intraepithelial neoplasia grade 1 biopsy. Obstet Gynecol. 2011 Dec;118(6):1222-1229. doi: 10.1097/AOG.0b013e318237caf4.

https://pubmed.ncbi.nlm.nih.gov/22105250/

Sørbye SW, Arbyn M, Fismen S, Gutteberg TJ, Mortensen ES. HPV E6/E7 mRNA testing is more specific than cytology in post-colposcopy follow-up of women with negative cervical biopsy. PLoS One. 2011;6(10):e26022. doi: 10.1371/journal.pone.0026022.

https://pubmed.ncbi.nlm.nih.gov/21998748/

Reviewer 2 Report

Due July 5, 2021

Review request: Biology

Manuscript ID: biology-1258707

Title:    The influence of sexual behavior and demographic characteristics in the expression of HPV-related biomarkers in a colposcopy population of reproductive age Greek women

Synopsis

The prospective observational study with multiple sources of recruitment involving patients who were referred due to abnormal gynecological findings (lab) was reported. The data from October 2016 to June 2017 were analyzed. “Greek women with referral abnormal cytology of any grade as well as those who tested HPV DNA and/or mRNA E6&E7 positive underwent colposcopic evaluation.” (Line 130-1) The baseline patient characteristics including referred diagnostic results are summarized in Table 1. A bar graph (Figure 1) shows the distribution of HPV subtypes. The chi-square tests of the HPV-DNA positive versus negative and other variables are indicated in Table 2. Figure 2 shows the HPV-DNA negative odds for five characteristics: coitarche, number of sexual partners, smoking, change of sexual partner in the past year, and HPV vaccine status. Figure 3 describes the interaction effect of coitarche or number of sexual partners versus smoking or HPV vaccine status. The chi-square tests of the HPV mRNA positive versus negative and other variables are indicated in Table 3.  Table 4 and Table 5 summarize the cut-off validity of diagnostic tests using the Papanicolaou smear or colposcopy as the gold standard, respectively. Combined lifestyle scoring systems and laboratory-based biomarker tests can impact the development of more valid HPV-related cervical pre-cancer risk assessment.

Reviewer's conflict of interest: None

General comment:

  1. Itis “a gold standard” instead of “golden standard” (Lines 80, 330, and 339)

Specific comments are listed below:

  1. Title: A period at the end of a title is not needed.
  2. Abstract: Either in the Simple Summary section or in the Abstract indicate the data collection range: October 2016 – June 2017.
  3. Introduction: no comment
  4. Methods and Materials:
  5. Line 92: Rather than “who accepted to participate” to the study,”, who signed the informed consent form” may be more appropriate.
  6. Line 97: “Constriction” refers to physical narrowing or a knot such as for a bore, traffic, or ponytail. Also the HPV immunization status is an independent variable. Therefore, “There was no constriction regarding the anti-HPV vaccination status in terms of the eligibility to the study” can be eliminated.
  7. Line 110: “Cervical Intraepithelial Neoplasia (CIN)” should be spelled out. Smoking as well as oral contraception may be factors affecting HPV and CIN status.
  8. Line 135: “LLETZ conization is considered the gold standard for histological diagnosis since it includes the entirety of the cervical transformation zone and was preferred over multiple punch biopsies in women who had completed childbearing” can be cited with PMID: 31430562 that supports it as a better diagnostic procedure.
  9. Section 2.3 Statistical analysis. AUC-ROC is the most important evaluation metric for checking on classification model performance. I wish you would mention the justification for using the Yuden index to determine the optimum cutoff value.
  10. Results:
    1. This section contains lots of information. The indication of data availability and data sharing procedures will be appreciated because more analyses may yield interesting secondary findings.
    2. Figure 2 can use a caption or footnote to clarify that the ORs are the reciprocal of the ORs for positive test results. The higher the OR, the more likely you are to have a negative on the HPV-DNA test. The lower the OR, the more likely you are to have a positive result. Reciprocal ORs may be more intuitive in this figure.
    3. For Figure 3, have you analyzed the interaction between two significant factors such as coitarche and number of sex partners?
  11. Discussion:

Line 358: What does HG stand for? Hyperemesis Gravidarum does not fit here. Please make sure all acronyms are indexed.

  1. Please add the Data Sharing Policy after the conclusion.

End of review.

Author Response

List of changes

Reviewer 2

Synopsis

The prospective observational study with multiple sources of recruitment involving patients who were referred due to abnormal gynecological findings (lab) was reported. The data from October 2016 to June 2017 were analyzed. “Greek women with referral abnormal cytology of any grade as well as those who tested HPV DNA and/or mRNA E6&E7 positive underwent colposcopic evaluation.” (Line 130-1) The baseline patient characteristics including referred diagnostic results are summarized in Table 1. A bar graph (Figure 1) shows the distribution of HPV subtypes. The chi-square tests of the HPV-DNA positive versus negative and other variables are indicated in Table 2. Figure 2 shows the HPV-DNA negative odds for five characteristics: coitarche, number of sexual partners, smoking, change of sexual partner in the past year, and HPV vaccine status. Figure 3 describes the interaction effect of coitarche or number of sexual partners versus smoking or HPV vaccine status. The chi-square tests of the HPV mRNA positive versus negative and other variables are indicated in Table 3.  Table 4 and Table 5 summarize the cut-off validity of diagnostic tests using the Papanicolaou smear or colposcopy as the gold standard, respectively. Combined lifestyle scoring systems and laboratory-based biomarker tests can impact the development of more valid HPV-related cervical pre-cancer risk assessment.

Reviewer's conflict of interest: None

General comment: Itis “a gold standard” instead of “golden standard” (Lines 80, 330, and 339)

Authors’ actions: Thank you for addressing this incorrect phrase which has been amended in the revised form.

Specific comments are listed below:

Comment 1: Title: A period at the end of a title is not needed.

Authors’ actions: Thank you for your point; this has been corrected in the revised form.

Comment 2: Abstract: Either in the Simple Summary section or in the Abstract indicate the data collection range: October 2016 – June 2017.

Authors’ actions: Thank you for your point; this has been amended in the revised form.

Comment 4: Line 92: Rather than “who accepted to participate” to the study,”, who signed the informed consent form” may be more appropriate.

Authors’ actions: Thank you for your point; your suggestion is indeed more appropriate and has been altered accordingly in the revised form.

Comment 5: Line 97: “Constriction” refers to physical narrowing or a knot such as for a bore, traffic, or ponytail. Also the HPV immunization status is an independent variable. Therefore, “There was no constriction regarding the anti-HPV vaccination status in terms of the eligibility to the study” can be eliminated.

Authors’ actions: Thank your point; this word has been changed to “prerequisites” in the revised form.

Comment 6: Line 110: “Cervical Intraepithelial Neoplasia (CIN)” should be spelled out. Smoking as well as oral contraception may be factors affecting HPV and CIN status.

Authors’ actions: Thank you for your remark. CIN was spelled out in the revised manuscript, the interrelation of smoking and oral contraceptives with both HPV persistence and CIN development predominantly though the pathways of atypical immature squamous metaplasia and CIN are well established.

Comment 7: Line 135: “LLETZ conization is considered the gold standard for histological diagnosis since it includes the entirety of the cervical transformation zone and was preferred over multiple punch biopsies in women who had completed childbearing” can be cited with PMID: 31430562 that supports it as a better diagnostic procedure.

Authorsactions: Thank you, we have acted accordingly.

Comment 8: Section 2.3 Statistical analysis. AUC-ROC is the most important evaluation metric for checking on classification model performance. I wish you would mention the justification for using the Yuden index to determine the optimum cut-off value.

Authors’ actions: Thank you for your comment. 1) We applied the proposed methodology for the coitarche and the number of sex partners, in the revised version the AUC of the ROC curves along with the 95% confidence intervals are reported, both for DNA positivity and for mRNA positivity. Moreover, we added a short description in section 2.3. 2) We used the Youden index as a single numeric metric to show a combined performance on the basis of sensitivity and specificity, however it seems that it was more confusing than highlighting! In the revised version we do not use this index any more, we consider the diagnostic odds ratio that is already reported and the overall accuracy sufficient.

Comment 9: Results: This section contains lots of information. The indication of data availability and data sharing procedures will be appreciated because more analyses may yield interesting secondary findings.

Authors’ actions: Thank you for your remark. Data are available from the corresponding author upon a reasonable request.

Comment 10: Results: Figure 2 can use a caption or footnote to clarify that the ORs are the reciprocal of the ORs for positive test results. The higher the OR, the more likely you are to have a negative on the HPV-DNA test. The lower the OR, the more likely you are to have a positive result. Reciprocal ORs may be more intuitive in this figure.

Authors’ actions: Thank you for your comment. Such a note is added in the figure caption, thank you.

Comment 11: Results: For Figure 3, have you analyzed the interaction between two significant factors such as coitarche and number of sex partners?

Authors’ actions: Thanks for pointing, we tested using the correlation coefficient and added the following information in the manuscript: “Note that the Spearman correlation coefficient between the age at first sexual intercourse and number of sexual partners was rs = -0.05 (p=0.3969) indicative that these two characteristics were not related.”, moreover we added a few sentences in the statistical methodology for completeness.

Comment 12: Discussion: Line 358: What does HG stand for? Hyperemesis Gravidarum does not fit here. Please make sure all acronyms are indexed.

Authors’ actions: We appreciate your point, HG stands for High Grade histology; we have double-checked that all acronyms are indexed.

Comment 13: Please add the Data Sharing Policy after the conclusion 

Authors’ actions: Thank you for your remark. Data are available from the corresponding author upon a reasonable request.

Round 2

Reviewer 1 Report

All comments have been addressed. The response to the comments is adequate and improved the manuscript.